# Evaluation of Renal Blood Flow in Dogs during Short-Term Human-Dose Epoprostenol Administration Using Pulsed Doppler and Contrast-Enhanced Ultrasonography

**DOI:** 10.3390/ani12091175

**Published:** 2022-05-04

**Authors:** Kiwamu Hanazono, Takaharu Itami, Ikuto Hayasaka, Kenjiro Miyoshi, Ai Hori, Keiko Kato, Daiji Endoh

**Affiliations:** School of Veterinary Medicine, Rakuno Gakuen University, 582 Bunkyodai Midorimach, Ebetsu 069-8501, Hokkaido, Japan; musashi1910@icloud.com (I.H.); v-drmys@rakuno.ac.jp (K.M.); hori@rakuno.ac.jp (A.H.); k-kato@rakuno.ac.jp (K.K.); dendoh@rakuno.ac.jp (D.E.)

**Keywords:** anesthesia, chronic kidney disease, contrast-enhanced ultrasonography, dog, epoprostenol, prostacyclin, pulsed Doppler ultrasonography, renal circulation

## Abstract

**Simple Summary:**

Since there is a lack of information regarding how epoprostenol, a prostacyclin, affects canine renal blood flow (RBF), we investigated the effects of short-term administration of epoprostenol at human doses of 2, 5, and 10 ng/kg/min intravenously for 20 min on RBF in six healthy dogs under anesthesia. The effects of short-term administration were investigated. As the dose of epoprostenol increased, peak systolic and end diastolic velocities of the renal arteries, maximum and minimum venous flow velocities of the interlobular and renal veins, and heart rate all tended to increase. However, these increases were not significant. These results indicate that the administration of human doses of epoprostenol to dogs does not produce significant changes in renal or systemic circulation.

**Abstract:**

Prostacyclin is an in vivo bioactive substance that regulates renal blood flow (RBF). Information regarding how epoprostenol, a prostacyclin preparation, affects RBF in dogs is lacking. We investigated the effects of short-term epoprostenol administration on RBF in six healthy dogs under anesthesia by administering it intravenously at human doses—2, 5, and 10 ng/kg/min for 20 min. RBF was evaluated before and during epoprostenol administration using pulsed Doppler ultrasonography, and renal perfusion was evaluated using contrast-enhanced ultrasonography. Effects on renal and systemic circulation were evaluated by measuring systolic arterial, mean arterial, diastolic arterial, pulmonary arterial, mean right atrial, and pulmonary capillary wedge pressures; heart rate; and cardiac output. Kruskal–Wallis and Bonferroni multiple comparison tests and Spearman’s rank correlation coefficient were used for statistical analyses. As epoprostenol dosage increased, the peak systolic and end diastolic velocity of the renal artery, maximum and minimum venous flow velocities of the interlobular and renal veins, and heart rate all tended to increase, although not significantly. Our results indicate that human-dose epoprostenol administration in dogs does not cause significant changes in renal or systemic circulation. However, the human doses used may have been too low to produce a clinical effect in dogs.

## 1. Introduction

Chronic kidney disease (CKD) is fatal and causes a persistent and irreversible decline in renal function, which can lead to uremia [1,2]. In patients with CKD, further rapid decline in kidney function can occur, which is termed Acute on Chronic Kidney Disease (ACKD) [3]. Recently, there has been increasing attention on the relationship between CKD progression and interstitial lesions, and it has been reported that interstitial fibrosis, in particular, leads to CKD exacerbation and is closely related to its progression [4,5]. Moreover, ACKD is being studied in both human and veterinary medicine to identify the risk factors that could contribute to CKD progression [3,6,7]. In the long-term management of renal failure, inhibiting interstitial fibrosis and increasing renal blood flow may be an effective treatment to slow CKD progression and further improve the prognosis of ACKD.

Prostacyclin is an in vivo biologically active substance that regulates renal blood flow. Conventionally, the main effects of prostacyclin are vasodilation, inhibition of platelet aggregation, interstitial fibrosis inhibition, and endothelial protection. Prostacyclin exerts its vasodilatory and anticoagulant effects by activating prostacyclin receptors (IP receptors) present on vascular smooth muscle cells and platelets, which consequently activates adenylate cyclase and increases cyclic adenosine monophosphate levels. Moreover, prostacyclin reportedly suppresses inflammation and fibrosis in the renal interstitium via IP receptors and exerts nephroprotective effects [8,9]. Furthermore, renal vasodilatation and nephroprotective effects have been reported in dogs [10,11,12].

Epoprostenol is a biosynthesized form of prostacyclin and is available for intravenous administration. Epoprostenol has been used in humans for the treatment of pulmonary arterial hypertension. Generally, the epoprostenol dosage for pulmonary hypertension in humans is very low, ranging from 2 to 10 ng/kg/min [13]. However, a study in dogs reported that epoprostenol did not have a sufficient pulmonary vasodilator effect even at a dose of 10 ng/kg/min [14], suggesting that dogs require a higher dose of epoprostenol than humans. However, higher doses reportedly affect systemic circulation, and could cause systemic hypotension [15].

Ultrasonography and, sometimes, pulsed Doppler ultrasonography and contrast-enhanced ultrasonography are non-invasive methods used for assessing renal circulation. Pulsed Doppler ultrasonography has been studied in humans since the 1990s, as a non-invasive and relatively simple method for assessing renal blood flow, using the pulsatility index (PI), resistive index (RI), and venous impedance index (VII) [16,17]. RI and PI have also been used in veterinary medicine to assess renal blood flow [18,19,20,21,22,23]. Contrast-enhanced ultrasonography has recently been studied as a renal perfusion assessment method, and there have been reports evaluating renal perfusion in both healthy and diseased patients in veterinary and human medicine [24,25,26,27,28].

Based on the hypothesis that epoprostenol alters renal blood flow by inducing vasodilatation, we investigated the effects of epoprostenol on renal and systemic circulation by intravenously administering incremental doses of epoprostenol to healthy dogs under anesthesia, in doses similar to those used for pulmonary hypertension in humans, and evaluated renal circulation using ultrasonography.

## 2. Materials and Methods

### 2.1. Animals

Six healthy beagle dogs (three male and three female dogs; age, 4–6 years; weight, 9.2–15.4 kg) were used in this study. All dogs were housed in the same kennel and were confirmed to be healthy by veterinarians through preliminary physical examinations, complete blood count evaluations, and blood chemistry tests. This prospective study was designed according to the 3R (Replacement, Reduction, and Refinement) Principle and was approved by the Rakuno Gakuen University Animal Experiment Committee (Approval number: VH19B15).

### 2.2. Anesthesia

Each dog was fasted for 12 h prior to anesthesia induction. Anesthesia was induced by placing a 22 G indwelling needle into the radial cutaneous vein and administering 6 mg/kg of propofol (PropoFlo, DS Pharma Animal Health, Osaka, Japan). After tracheal intubation, anesthesia was maintained using sevoflurane (SevoFlo, Maruishi Pharmaceutical, Osaka, Japan) at a concentration of 1.2–2.5%. Anesthesia was managed so that the mean arterial blood pressure (MABP) was above 70 mmHg and the body temperature was above 37.0 °C. Following the placement of a pulmonary artery catheter, 25 mg/kg of cefazolin sodium (Cefamezin α, LTL Pharma, Tokyo, Japan) was administered intravenously during anesthesia induction, and 0.2 mg/kg of meloxicam (Metacam, Boehringer Ingelheim Animal Health Japan, Tokyo, Japan) was administered upon awakening; 2 mg/kg of the latter was also administered subcutaneously. During anesthesia, 1 mg/kg/h of rocuronium bromide (ESLAX, MSD Pharmaceuticals, Tokyo, Japan) was administered by continuous quantitative drip infusion to prevent spontaneous respiration in the dogs. After completing all required measurements, 4 mg/kg of sugammadex sodium (Bridion, MSD, Tokyo, Japan) was administered intravenously, and the dogs were awakened promptly after sufficient spontaneous respiration was confirmed.

### 2.3. Epoprostenol Administration

Epoprostenol (Epoprostenol “ACT”, Actelion Pharmaceuticals Japan, Tokyo, Japan) was diluted in physiological saline and administered at 2, 5, and 10 ng/kg/min, with a graduated increase in dose over time, through the 22 G indwelling needle in the radial cutaneous vein. Various interventions, such as blood pressure measurement and ultrasonography, were performed after a stabilization period of approximately 20 min following the initial epoprostenol administration at the default dose. During anesthesia, infusion was performed using lactated Ringer’s solution (SOLULACT F, Terumo, Tokyo, Japan). The infusion volume was adjusted such that the sum of the rocuronium and epoprostenol infusion rates was always 5 mL/kg/h.

### 2.4. Measurements

After anesthesia induction, an electrocardiography was performed and a catheter was placed in the right dorsal leg artery to monitor heart rate (HR), systolic arterial blood pressure (SABP), MABP, and diastolic arterial blood pressure (DABP). A pulmonary artery catheter (Swan Ganz Thermodilution Catheter, Edwards Lifesciences, Tokyo, Japan) and a pressure transducer (DS-7210, Fukuda Denshi, Tokyo, Japan) were placed in the left jugular vein, and mean pulmonary artery pressure (mPAP), mean right atrial pressure (mRAP), and pulmonary capillary wedge pressure (PCWP) were measured. Moreover, cardiac output (CO) was measured using thermodilution. For thermodilution, a 5% glucose solution (Glucose Note 5%, TERUMO, Tokyo, Japan) cooled to approximately 5 °C was used. Each measurement was performed at four time points: before starting epoprostenol administration (pre-administration), 20 min after commencing the 2 ng/kg/min dose (2 ng/kg/min), 20 min after commencing the 5 ng/kg/min dose (5 ng/kg/min), and 20 min after commencing the 10 ng/kg/min dose (10 ng/kg/min). The first measurement was performed after a stabilization period of at least 60 min after the pulmonary artery catheter was placed to avoid the effects of mechanical stimulation associated with catheter puncture. The above measurements were generally in accordance with the methods used in our previous studies [29,30].

A diagnostic ultrasound system (ALIETTA70, Hitachi, Tokyo, Japan) was used for ultrasonographic guidance. After anesthesia induction, the dogs were kept in the right lateral recumbent position. Only the left kidney was delineated using a convex probe (C42, Hitachi, Tokyo, Japan), and color Doppler ultrasonography was used to delineate the interlobular artery and interlobular vein in horizontal sections, as well as the renal artery and renal vein in transverse sections. Pulsed Doppler ultrasonography (sample volume 1.0–6.0 mm, 8 MHz) was used to record the blood flow waveforms of the interlobular artery, interlobular vein, renal artery, and renal vein at the same location three times each (Figure 1). In all measurements, the Doppler angle of incidence was within 60°, and the correction angle was within 30°. Ultrasonography was performed at four time points: pre-administration and at administration of 2, 5, and 10 ng/kg/min doses.

PI, RI, peak systolic velocity (PSV), end diastolic velocity (EDV), and time average maximum velocity (TAMV) were determined from the interlobular and renal artery waveforms (Figure 1). The maximum (Vmax) and minimum (Vmin) venous flow velocities were measured from the blood flow waveforms of the interlobar and renal veins, respectively (Figure 1). The VII was calculated from the Vmax and Vmin of the interlobar and renal veins using the formula described below. The mean of all three measurements was used for statistical analysis. Each calculation was performed using the ultrasound system described previously.

The three measurements were calculated using the following equations:PI = (PSV − EDV)/TAMV
RI = (PSV − EDV)/PSV
VII = (Vmax − Vmin)/Vmax

The aforementioned ultrasound system was used for contrast-enhanced ultrasonography; a 7-MHz linear probe (L34, Hitachi, Tokyo, Japan) was used to delineate the long-axis section of the left kidney, followed by intravenous administration of 0.01 mL/kg of a contrast agent for ultrasonography (Sonazoid, Daiichi Sankyo, Tokyo, Japan) and a subsequent boosting dose of 2.0 mL of saline. Images were recorded up 120 s after contrast administration. Contrast-enhanced ultrasonography was performed once at each of the four time points: pre-administration and at administration of 2, 5, and 10 ng/kg/min doses.

Time intensity curves (TICs) were plotted, based on the recorded contrast-enhanced ultrasound images, using the contrast harmonic imaging (CHI) analysis function of the ultrasound system. The region of interest (ROI) was set as a 2.0 × 2.0-mm circle in both the cortical and medullary regions of the left kidney (Figure 2).

From the generated TICs, time-related parameters (reflecting the time taken for the passage of the contrast medium) and intensity-related parameters (reflecting the contrast intensity) were measured. The time-related parameters included time-to-peak (TTP; the time from the start of contrast administration to the maximum intensity), rising time (RT; the time from the start of contrast enhancement to maximum intensity), wash-in Ratio (WiR; the rate at which the contrast medium flows in), and wash-out ratio (WoR; the rate at which the contrast medium flows out). The following intensity-related parameters were measured: base intensity (BI; the intensity before the start of contrast enhancement), peak effect (PE; maximum intensity), overall area under the curve (AUC), wash-in AUC (WiAUC; AUC during contrast enhancement), and wash-out AUC (WoAUC; AUC during contrast attenuation) (Figure 3).

### 2.5. Statistical Analyses

The Kruskal–Wallis and Bonferroni’s multiple comparison tests were used to examine significant differences in systemic circulation parameters, renal blood flow parameters on pulsed Doppler ultrasonography, and renal perfusion parameters of contrast-enhanced ultrasonography at each of the time points. In addition, Spearman’s rank correlation coefficient was used to examine the correlation between systemic circulation parameters and renal perfusion parameters. A *p*-value < 0.05 was considered statistically significant. R (R Foundation for Statistical Computing, Vienna, Austria), a free language and environment for statistical computing, was used for statistical analyses.

## 3. Results

### 3.1. Pulsed Doppler Ultrasonography

Table 1 shows the measured values between the pre-administration and 10 ng/kg/min administration time points. PSV (renal artery), EDV (renal artery), Vmax (both interlobular vein and renal vein), and Vmin (both interlobular vein and renal vein) increased from pre-administration levels to the 10 ng/kg/min time point. No significant differences were observed between time points.

### 3.2. Contrast-Enhanced Ultrasonography

Table 2 shows the measured values from the pre-administration levels to the 10 ng/kg/min administration time point. Cortical TTP and cortical AUC decreased from the 2 ng/kg/min to the 10 ng/kg/min time point. Medullary WiR and medullary PE increased from the 2 ng/kg/min to the 10 ng/kg/min time point. No significant differences were observed between the time points.

### 3.3. Systemic Circulation Parameters

The measured values of the systemic circulatory parameters, from pre-administration to the 10 ng/kg/min time point, are shown in Table 3. The HR increased from the pre-administration to the 10 ng/kg/min time point. However, there were no significant differences in any of the parameters between the time points.

The correlation between each parameter measured by ultrasonography and systemic circulation parameters is shown in Table 4. Negative correlations between the PI and RI of the interlobular artery and the MABP (r = −0.67 and r = 0.7; *p* < 0.01), DABP (r = −0.82 and r = −0.81, *p* < 0.01, respectively), HR (r = −0.67 and r = −0.63, *p* < 0.01, respectively), and CO (r = −0.41 and r = −0.43, *p* < 0.05, respectively) were observed. The EDV of the interlobular artery was positively correlated with the MABP (r = −0.46, *p* = 0.03) and negatively correlated with the DABP (r = −0.52, *p* < 0.01). There was a positive correlation between the Vmax of the interlobular vein and the HR (r = 0.58, *p* < 0.01) and CO (r = 0.45, *p* = 0.03). The Vmin of the interlobular vein was negatively correlated with the DABP (r = −0.41, *p* < 0.04) and positively correlated with the HR (r = 0.58, *p* < 0.01) and CO (r = 0.51, *p* = 0.01). There was a negative correlation between the VII of the interlobular vein and the HR (r = −0.43, *p* = 0.04).

There was a negative correlation between the PI and RI of the renal artery and the MABP (r = −0.51 and r = −0.50, *p* = 0.01, respectively), DABP (r = −0.73, *p* < 0.01, both), HR (r = −0.51, *p* = 0.01; and r = −0.49, *p* = 0.02, respectively), and CO (r = −0.45, *p* = 0.03; and r = −0.47, *p* = 0.02, respectively). There was a positive correlation between the PSV of the renal artery and the SABP (r = 0.60, *p* < 0.01). There was a positive correlation between the EDV of the renal artery and the MABP (r = 0.55, *p* < 0.01), DABP (r = 0.60, *p* < 0.01), HR (r = 0.41, *p* = 0.048), and CO (r = 0.63, *p* < 0.01). There was a positive correlation between the Vmax of the renal vein and the MABP (r = 0.63, *p* < 0.01), DABP (r = 0.55, *p* < 0.01), HR (r = 0.57, *p* < 0.01), and CO (r = 0.59, *p* < 0.01). There was a positive correlation between the Vmin of the renal vein and the MABP (r = 0.66, *p* < 0.01), DABP (r = 0.59, *p* < 0.01), HR (r = 0.60, *p* < 0.01), and CO (r = 0.64, *p* < 0.01). There was a negative correlation between the VII of the renal vein and the MABP (r = −0.44, *p* = 0.03), DABP (r = −0.48, *p* = 0.02), and CO (r = −0.43, *p* = 0.04).

The correlation between each parameter measured using contrast-enhanced ultrasonography and systemic circulation parameters is shown in Table 4. Regarding the correlation between cortical parameters and systemic circulation parameters, the HR was positively correlated with the cortical PE (r = 0.58, *p* < 0.01), BI (r = 0.46, *p* = 0.02) and WiR (r = 0.58, *p* < 0.01), and was negatively correlated with the cortical TTP (r = −0.61, *p* < 0.01) and WiAUC (r = −0.44, *p* = 0.03).

The CO was positively correlated with the cortical WoAUC (r = 0.44, *p* = 0.03) and negatively correlated with the cortical WoR (r = −0.52, *p* < 0.01). The DABP was negatively correlated with the cortical WiAUC (r = −0.54, *p* < 0.01).

Regarding the correlation between medullary parameters and systemic circulation parameters, the HR was positively correlated with the medullary PE (r = 0.49, *p* = 0.02) and negatively correlated with the medullary TTP (r = −0.69, *p* < 0.01); CO was positively correlated with the medullary AUC (r = 0.43, *p* = 0.04).

In one of the dogs that received epoprostenol, bleeding was observed at the insertion site of the sheath postoperatively; however, the bleeding was stopped by compression hemostasis. Another dog had soft stools postoperatively; however, there was no soft stool the following day.

## 4. Discussion

Veraprost sodium, an oral prostacyclin derivative, has been used for CKD treatment in cats and has been reported to reduce renal function decline and improve clinical signs in cats with International Veterinary Nephrology Study Group stage 2–3 CKD [31]. Although the usefulness of prostacyclin preparations for CKD has been established, there are no reports on the effects of epoprostenol, an injectable prostacyclin, on renal circulation at human doses in dogs.

The results of the present study using pulsed Doppler ultrasonography showed an upward trend in the PSV and EDV of the interlobular and renal arteries and the Vmax and Vmin of the renal veins from the pre-administration to the 10 ng/kg/min administration time points. We believe that these results are related to the increase in HR from the pre-administration to the 10 ng/kg/min time points, as described below. Although there were changes in the measured values for some items, these evaluations were all subjective, and significant differences were not detected in the measured values of all items when going from the pre-administration to the 10 ng/kg/min administration time points. These results suggest that the epoprostenol dose in this study had no obvious effects on blood flow in the interlobular and renal arteries and veins.

Contrast-enhanced ultrasonography is an indicator of renal function in humans. In CKD progression, there is a decrease in PE and WiR, and an increase in WoR in the renal cortex; moreover, there is a significant correlation between PE, WiR, WoR, and eGFR [32]. In canine renal contrast ultrasonography, delayed time-related parameters were reported in a chronic ischemic kidney disease model [28]. In this study, cortical TTP decreased from the 2 to the 10 ng/kg/min administration points, but significant differences were not detected in time-related parameters. Thus, the epoprostenol dose in this study had no apparent effect on renal perfusion.

After starting epoprostenol administration, not only renal circulation but also systemic circulation parameters were expected to be affected, and hypotension and bradycardia were mainly considered [10]. In our study, there was no significant change in pulmonary artery pressure or any other systemic circulatory parameters, but the HR increased and the right atrial pressure decreased from the pre-administration to the 10 ng/kg/min time points. This increase in HR and CO and decrease in blood pressure could have been due to two factors: (1) the decrease in blood pressure due to prolonged anesthesia, and (2) the decrease in blood pressure due to the vasodilator effect of epoprostenol. However, these causes were not identified in this study. Moreover, as with the assessment of renal blood flow, these changes were based on subjective assessments, and significant differences were not detected between measurements at each time point from the pre-administration to the 10 ng/kg/min time points. This possibly indicates that epoprostenol administration in this study had no obvious effect on systemic circulation.

In this study, each item in the pulsed Doppler method was significantly correlated with systemic circulation parameters. In contrast ultrasonography, significant correlations were also detected between renal perfusion parameters (such as PE, TTP, WiR, and WoR) and systemic circulation parameters, suggesting that HR and CO are closely related to renal perfusion. The kidney is a perfused organ that is strongly influenced by systemic circulation [33,34]. However, only a few reports have evaluated the correlation between renal and systemic circulation parameters using pulsed Doppler ultrasonography and contrast-enhanced ultrasonography in dogs and cats; therefore, the results of this study may provide useful information.

In this study, we found that epoprostenol had no apparent effect on renal blood flow and systemic circulation; however, since the nephroprotective effect of veraprost sodium (which also acts as a prostacyclin) has been reported [31,35], we cannot exclude the possibility that epoprostenol may have had a similar effect. There are three main reasons for the lack of obvious epoprostenol effects on renal blood flow in this study.

First, epoprostenol doses (2–10 ng/kg/min) in this study were based on human doses and may not have been effective enough in dogs. In a previous study, when prostacyclin was administered via the renal artery in five dogs, the dose in renal blood flow was increased from 30 to 300 ng/min over a period of 15–30 min, and adverse effects, such as hypotension and bradycardia, were observed at doses of 1000–3000 ng/min. [10]. However, in the aforementioned study, prostacyclin was administered directly into the renal artery, unlike the method used in this study. The results of this study suggest that epoprostenol may need to be administered at higher doses in dogs than those in humans to increase renal blood flow when administered intravenously.

Second, the observational period in this study was short. In addition to vasodilatation, prostacyclin also inhibits platelet aggregation, protects the vascular endothelium, and inhibits interstitial fibrosis, which are thought to exert reno-protective effects in the long term. When prostacyclin is infused directly into the renal artery in dogs, renal blood flow increases immediately after commencing administration, which is thought to be mainly due to the vasodilator effect of prostacyclin [10,36,37]. In contrast, a study investigating the usefulness of veraprost sodium for CKD in cats showed its usefulness after a long observation period of 180 days [31], and many clinical studies of prostacyclin in humans have been conducted over a long observation period [38]. This suggests that the inhibition of platelet aggregation, protection of the vascular endothelium, and inhibition of interstitial fibrosis (which are effects of prostacyclin) are also involved in its long-term reno-protective effects [35]. Therefore, monitoring RBF parameters over a longer period will confirm the usefulness of epoprostenol.

Third, this study was conducted under anesthesia. It has been reported that anesthesia or sedation decreases blood pressure and HR, resulting in increased PI and RI, and prolonged TTP [34,39]. Therefore, in this study, we cannot deny the possibility that changes in RBF parameters induced by epoprostenol may have been masked by changes in systemic circulation caused by anesthesia. In the future, we may be able to observe the effects of epoprostenol on RBF more clearly by assessing it without anesthesia.

Another limitation is that only a small number of dogs were assessed; therefore, evaluating a larger number of animals is necessary in the future. Moreover, evaluating cats (which are frequently affected by chronic kidney disease) or evaluating diseased animals would provide more useful information.

## 5. Conclusions

In conclusion, epoprostenol administration at human doses (2–10 ng/kg/min) and short-term monitoring in dogs did not clearly result in altered renal blood flow or systemic circulation effects due to the vasodilatory effect of prostacyclin. Our results indicate that human-dose epoprostenol administration in dogs does not cause significant changes in renal or systemic circulation. The doses used in this study were based on human doses, and it is possible that the doses were too low to produce a clinical effect in dogs.

## Figures and Tables

**Figure 1 animals-12-01175-f001:**
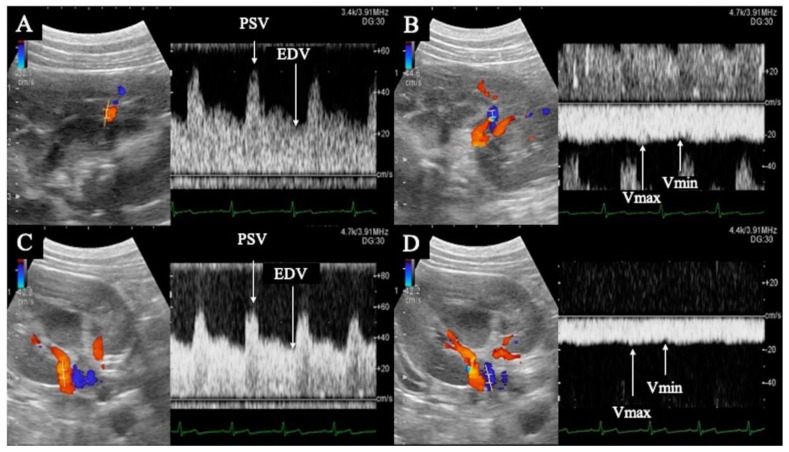
Flow waveform of the renal interlobar artery (**A**) and vein (**B**), and renal artery (**C**) and vein (**D**). (**A**,**B**) are horizontal sections of the kidney; (**C**,**D**) are transverse sections. The renal interlobar artery (**A**) and renal artery (**C**) show pulsatile waveforms with peak systolic velocity in systole (PSV) and end diastolic velocity (EDV) in diastole. The renal interlobar vein (**B**) and renal vein (**D**) show flat, continuous waveforms; the maximum velocity is Vmax and the minimum velocity is Vmin.

**Figure 2 animals-12-01175-f002:**
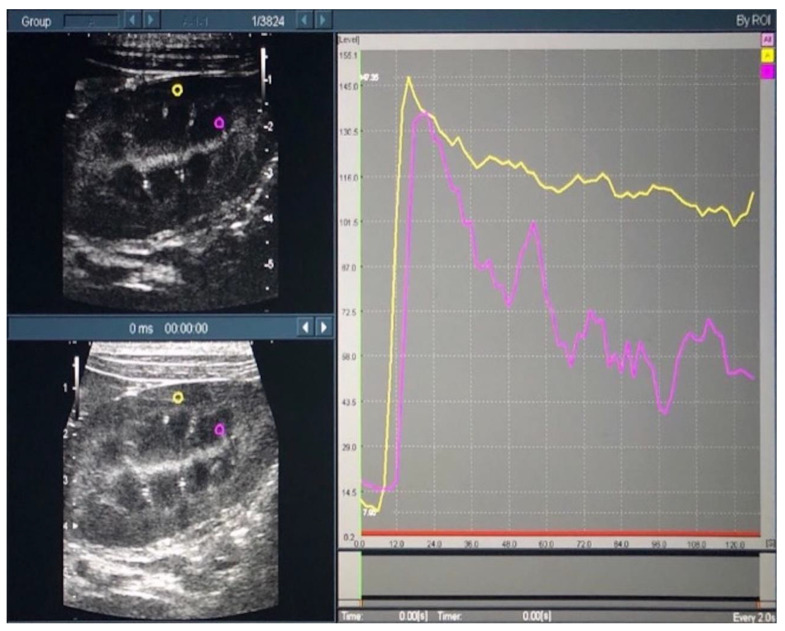
Time intensity curves of the renal cortex and medulla in contrast-enhanced ultrasonography. In the left figure, the yellow and purple circles indicate the regions of interest in the cortical and medullary regions, respectively. In the right figure, the yellow and purple lines indicate the time intensity curves in the cortical and medullary regions, respectively.

**Figure 3 animals-12-01175-f003:**
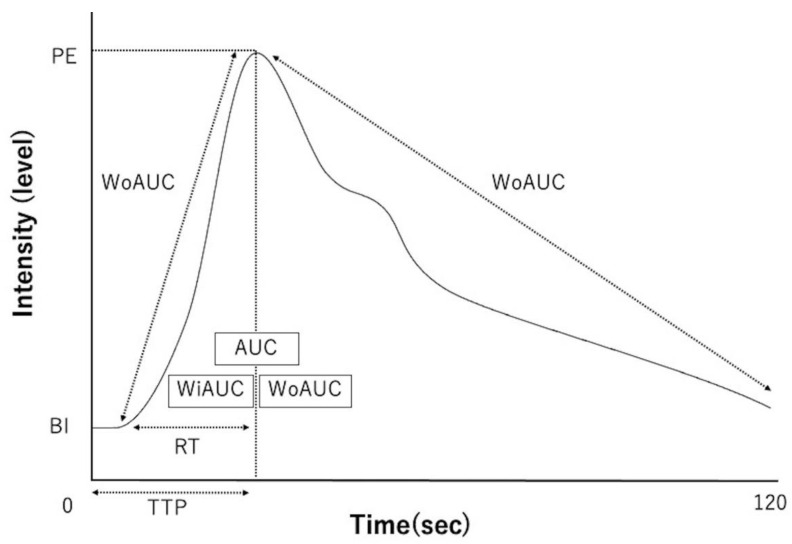
Schematic diagram of time intensity curves and each parameter. TTP—time-to-peak; RT—rising time; WiR—wash-in ratio; WoR—wash-out ratio; BI—base intensity; PE—peak effect (maximum intensity); AUC—area under the curve; WiAUC—wash-in AUC; WoAUC—wash-out AUC.

**Table 1 animals-12-01175-t001:** Comparison of renal blood flow parameters using the pulsed Doppler method at each time point.

Parts	Variables	Norms	Pre-Administration	2 ng/kg/min	5 ng/kg/min	10 ng/kg/min
Interlobular artery	PSV (cm/sec)	-	33.48 (18.83–52.07)	37.93 (13.57–53.33)	39.37 (20.77–52.33)	39.25 (27.87–49.10)
	EDV (cm/sec)	-	14.87 (7.40–23.83)	15.95 (5.47–28.70)	17.77 (9.13–20.40)	16.23 (8.57–24.33)
	PI	1.15 ± 0.15 [20]	0.99 (0.72–1.41)	0.90 (0.651.16)	0.93 (0.72–1.29)	0.96 (0.71–1.39)
	RI	0.62 ± 0.04 [20] 0.66 ± 0.05 [22]	0.60 (0.50–0.70)	0.55 (0.46–0.75)	0.57 (0.50–0.68)	0.58 (0.49–0.70)
Interlobular vein	Vmax (cm/sec)	-	12.48 (8.97–21.17)	12.92 (6.73–50.07)	13.75 (8.77–58.30)	16.78 (10.80–6.13)
	Vmin (cm/sec)	-	9.17 (6.83–16.27)	9.87 (4.87–40.00)	10.43 (6.47–48.43)	16.61 (7.90–38.03)
	VII	-	0.25 (0.21–0.31)	0.23 (0.20–0.38)	0.25 (0.17–0.29)	0.25 (0.18–0.39)
Renal artery	PSV (cm/sec)	75.2 ± 22.0 [23]	54.77 (35.80–74.30)	58.02 (46.80–91.07)	63.80 (48.20–70.10)	68.67 (43.40–111.40)
	EDV (cm/sec)	25.7 ± 8.2 [23]	25.23 (15.13–36.27)	27.90 (23.90–42.97)	29.83 (19.37–35.80)	30.42 (16.83–53.07)
	PI	1.34 ± 0.32 [23]	0.77 (0.65–1.26)	0.75 (0.60–1.25)	0.91 (0.57–1.38)	0.94 (0.57–1.28)
	RI	0.66 ± 0.07 [23]	0.50 (0.46–0.67)	0.50 (0.44–0.66)	0.56 (0.41–0.70)	0.56 (0.42–0.67)
Renal vein	Vmax (cm/sec)	-	14.28 (13.00–25.57)	16.95 (15.07–25.00)	19.27 (12.43–27.10)	20.32 (12.27–30.77)
	Vmin (cm/sec)	-	11.35 (9.83–21.10)	13.27 (9.70–19.70)	14.95 (6.73–22.27)	15.68 (6.77–24.47)
	VII	-	0.22 (0.17–0.28)	0.24 (0.19–0.36)	0.22 (0.18–0.46)	0.21 (0.20–0.45)

PSV—peak systolic velocity; EDV—end diastolic velocity; PI—pulsatility index; RI—resistive index; Vmax—maximum venous flow velocity; Vmin—minimum venous flow velocity; VII—venous impedance index. No significant differences were observed between time points.

**Table 2 animals-12-01175-t002:** Comparison of renal perfusion parameters using contrast-enhanced ultrasonography at each time point.

Parts	Variables	Pre-Administration	2 ng/kg/min	5 ng/kg/min	10 ng/kg/min
Renal cortex	TTP (sec)	13.10 (12.00–14.60)	13.80 (11.20–16.20)	13.20 (10.00–14.80)	12.60 (10.00–15.20)
	RT (sec)	4.30 (3.40–6.00)	4.55 (3.00–6.00)	4.90 (3.00–6.20)	4.80 (4.20–5.60)
	WiR	38.55 (26.00–64.80)	43.25 (31.40–76.80)	49.65 (29.20–60.60)	46.10 (30.70–81.40)
	WoR	0.60 (0.50–0.70)	0.60 (0.40–0.70)	0.55 (0.40–0.80)	0.60 (0.50–0.80)
	PE	139.55 (108.50–186.70)	155.55 (135.30–188.70)	156.25 (115.40–205.30)	155.65 (133.10–202.20)
	BI	10.50 (0.60–44.40)	15.75 (3.10–50.20)	11.95 (4.40–57.10)	12.95 (6.00–49.30)
	AUC	8451.30 (7468.80–11,105.90)	11,089. 10 (8263.00–12,154.50)	10,796.00 (5841.10–11,930.50)	9953.20 (7560.50–12,580.60)
	WiAUC	621.25 (386.20–845.60)	466.40 (383.30–717.40)	528.30 (445.60–724.70)	489.45 (388.60–726.50)
	WoAUC	7743.10 (7082.60–10,477.50)	10538.70 (7703.00–11,762.70)	10335.40 (5338.00–11,241.90)	9556.90 (6989.50–11,854.10)
Renal medulla	TTP (sec)	18.70 (16.60–22.80)	19.50 (15.20–25.20)	20.80 (15.20–25.20)	18.90 (15.20–25.00)
	RT (sec)	8.30 (6.80–9.20)	8.90 (5.60–12.40)	9.50 (5.00–13.40)	8.50 (7.40–11.80)
	WiR	21.85 (8.60–25.40)	14.25 (8.70–40.20)	16.50 (10.50–47.30)	17.75 (14.80– 31.60)
	WoR	0.75 (0.60–1.10)	0.80 (0.60–1.50)	1.00 (0.40–1.20)	0.85 (0.60–1.00)
	PE	138.20 (90.60–174.00)	132.05 (83.50–197.40)	137.80 (109.50–193.70)	138.75 (79.0–193.00)
	BI	16.10 (11.60–20.90)	14.60 (10.60–26.10)	18.65 (3.40–30.80)	18.35 (14.10–28.00)
	AUC	5560.60 (1861.80–7720.80)	3646.80 (1848.90–8293.00)	5235.80 (1890.90–8284.00)	4324.10 (946.90–9601.00)
	WiAUC	593.90 (255.80–917.20)	571.00 (347.80–1054.20)	635.20 (385.30–841.20)	491.60 (227.20–816.80)
	WoAUC	4891.9 (1606.00–7071.60)	2859.7 (1501.10–7238.90)	4489.40 (1505.60–7665.20)	3813.5 (719.70–8810.50)

TTP—time-to-peak; RT—rising time; WiR—wash-in ratio; WoR—wash-out ratio; BI—base intensity; PE—peak effect; AUC—area under the curve; WiAUC—wash-in AUC; WoAUC—wash-out AUC. No significant differences were observed between time points.

**Table 3 animals-12-01175-t003:** Comparison of systemic circulation parameters at each time point.

Variables	Norms	Pre-Administration	2 ng/kg/min	5 ng/kg/min	10 ng/kg/min
SABP (mmHg)	96.1 ± 3.5 [29]	118.53 (107.5–129.2)	123.85 (118.66–134.0)	121.54 (114.60–137.4)	124.25 (112.0–136.5)
MABP (mmHg)	68.5 ± 2.6 [29]	80.40 (75.20–85.00)	84.80 (79.66–93.00)	84.17 (82.25–93.20)	81.33 (78.75–95.00)
DABP (mmHg)	54.2 ± 2.0 [29]	67.98 (57.80–72.00)	71.83 (62.66–79.75)	68.85 (63.80–78.00)	66.30 (61.40–80.33)
mPAP (mmHg)	10.7 ± 0.2 [29]	11.63 (9.00–13.60)	12.83 (8.33–13.00)	12.97 (8.40–14.16)	12.50 (9.00–13.50)
mRAP (mmHg)	3.8 ± 0.3 [29]	3.50 (2.66–4.75)	4.00 (1.00–4.00)	3.00 (1.00–5.20)	2.50 (1.00–4.80)
PCWP (mmHg)	4.5 ± 0.2 [29]	5.25 (4.00–7.00)	5.75 (4.50–7.00)	5.25 (3.50–8.00)	5.00 (3.00–7.50)
HR (/min)	86.0 ± 3.9 [29]	98.50 (71.20–118.33)	105.50 (74.33–116.6)	113.07 (77.20–129.5)	119.88 (75.60–130.0)
CO (L/min)	1.8 ± 0.3 [30]	2.08 (1.30–2.66)	2.25 (1.20–2.48)	2.24 (1.49–2.44)	2.18 (1.48–2.59)

SABP—systolic arterial blood pressure; MABP—mean arterial blood pressure; DABP—diastolic arterial blood pressure; HR—heart rate; mPAP—mean pulmonary artery pressure; mRAP—mean right atrial pressure; PCWP—pulmonary capillary wedge pressure; CO—cardiac output. No significant differences were observed between time points.

**Table 4 animals-12-01175-t004:** Correlation between ultrasonographic parameters and systemic circulation parameters.

Parts	Variables	SABP		MABP		DABP		HR		CO	
		r	*p*	r	*p*	R	*p*	r	*p*	r	*p*
Interlobular artery	PSV	0.38	0.07	−0.01	0.98	−0.03	0.9	−0.16	0.44	−0.18	0.4
	EDV	0.33	0.12	**0.46**	**0.03**	**0.52**	**<0.01**	0.31	0.15	0.2	0.35
	PI	−0.15	0.49	**−0.67**	**<0.01**	**−0.82**	**<0.01**	**−0.67**	**<0.01**	**−0.41**	**0.046**
	RI	−0.14	0.5	**−0.7**	**<0.01**	**−0.81**	**<0.01**	**−0.63**	**<0.01**	**−0.43**	**0.04**
Interlobular vein	Vmax	0.05	0.82	0.25	0.23	**0.41**	**0.049**	**0.58**	**<0.01**	**0.45**	**0.03**
	Vmin	0.05	0.83	0.28	0.19	**0.41**	**0.04**	**0.58**	**<0.01**	**0.51**	**0.01**
	VII	−0.08	0.73	−0.29	0.16	−0.34	0.11	**−0.43**	**0.04**	−0.38	0.07
Renal artery	PSV	**0.6**	**<0.01**	0.12	0.57	−0.03	0.91	−0.13	0.55	0.18	0.41
	EDV	0.25	0.25	**0.55**	**<0.01**	**0.6**	**<0.01**	**0.41**	**0.048**	**0.63**	**<0.01**
	PI	0.16	0.45	**−0.51**	**0.01**	**−0.73**	**<0.01**	**−0.51**	**0.01**	**−0.45**	**0.03**
	RI	0.18	0.39	**−0.5**	**0.01**	**−0.73**	**<0.01**	**−0.49**	**0.02**	**−0.47**	**0.02**
Renal vein	Vmax	0.05	0.8	**0.63**	**<0.01**	**0.55**	**<0.01**	**0.57**	**<0.01**	**0.59**	**<0.01**
	Vmin	0.02	0.92	**0.66**	**<0.01**	**0.59**	**<0.01**	**0.60**	**<0.01**	**0.64**	**<0.01**
	VII	−0.09	0.67	**−0.44**	**0.03**	**−0.48**	**0.02**	−0.31	0.14	**−0.43**	**0.04**
Renal cortex	TTP	0.25	0.24	0.02	0.92	−0.27	0.21	**−0.61**	**<0.01**	−0.06	0.77
	RT	0.36	0.08	−0.05	0.8	−0.31	0.14	−0.35	0.09	0.05	0.83
	WiR	−0.22	0.31	0.25	0.24	0.33	0.12	**0.58**	**<0.01**	0.23	0.27
	WoR	0.35	0.1	−0.25	0.23	−0.21	0.33	−0.2	0.35	**−0.52**	**0.01**
	PE	−0.24	0.26	0.12	0.59	0.12	0.57	**0.58**	**<0.01**	0.15	0.5
	BI	0.04	0.84	0.22	0.3	0.27	0.21	**0.46**	**0.02**	−0.05	0.82
	AUC−	−0.28	0.19	0.27	0.2	0.15	0.48	0.31	0.14	0.39	0.06
	WiAUC	−0.32	0.12	−0.43	0.04	**−0.54**	**<0.01**	**−0.44**	**0.03**	−0.26	0.22
	WoAUC	−0.24	0.26	0.33	0.11	0.21	0.31	0.36	0.09	**0.44**	**0.03**
Renal medulla	TTP	0.29	0.18	−0.08	0.72	−0.36	0.08	**−0.69**	**<0.01**	−0.15	0.48
	RT−	0.09	0.69	0.16	0.47	−0.002	0.99	−0.29	0.17	0.15	0.47
	WiR	−0.28	0.18	−0.04	0.85	−0.11	0.62	0.4	0.05	0.16	0.46
	WoR	−0.08	0.72	0.19	0.38	0.22	0.29	0.38	0.07	−0.27	0.21
	PE	−0.39	0.06	0.02	0.92	0.05	0.83	**0.49**	**0.02**	0.11	0.6
	BI	0.21	0.32	0.07	0.75	0.15	0.48	0.3	0.15	0.05	0.8
	AUC	−0.25	−0.25	0.06	0.79	−0.02	0.94	0.36	0.08	**0.43**	**0.04**
	WiAUC	−0.23	0.28	0.33	0.12	0.14	0.5	0.09	0.66	0.28	0.19
	WoAUC	−0.28	0.18	0.001	0.99	−0.05	0.81	0.37	0.08	0.39	0.06

SABP—systolic arterial blood pressure; MABP—mean arterial blood pressure; DABP—diastolic arterial blood pressure; HR—heart rate; mPAP—mean pulmonary artery pressure; mRAP—mean right atrial pressure; PCWP—pulmonary capillary wedge pressure; CO—cardiac output; PSV—peak systolic velocity; EDV—end diastolic velocity; PI—pulsatility index; RI—resistive index; Vmax—maximum venous flow velocities; Vmin—minimum venous flow velocities; VII—venous impedance index; TTP—time-to-peak; RT—rising time; WiR—wash-in ratio; WoR—wash-out ratio; BI—base intensity; PE—peak effect; AUC—area under the curve; WiAUC—wash-in AUC; WoAUC—wash-out AUC. Values in bold indicate measurements with a significant correlation.

## Data Availability

Not applicable.

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
