# Peer review of "Evaluation of Renal Blood Flow in Dogs during Short-Term Human-Dose Epoprostenol Administration Using Pulsed Doppler and Contrast-Enhanced Ultrasonography"

_animals, 2022, doi:10.3390/ani12091175_

Round 1

Reviewer 1 Report

The manuscript is very interesting, described in a simple and clear way, as well as understandable to the reader.

Please add norms for healthy dogs in table 1,2 and 3. It is worth to refer the article "Evaluation of the diagnostic value of the renal resistive index as a marker of the subclinical development of cardiorenal syndrome in MMVD dogs" considering the norms for RI in healthy dogs and dogs with chronic renal failure and heart disease.

Author Response

<Reviewer 1>

・Please add norms for healthy dogs in table 1,2 and 3. It is worth to refer the article "Evaluation of the diagnostic value of the renal resistive index as a marker of the subclinical development of cardiorenal syndrome in MMVD dogs" considering the norms for RI in healthy dogs and dogs with chronic renal failure and heart disease.

Re1: Thank you for your suggestion. In line with suggestion, I have added reference values to Tables 1, 2, and 3 as much as possible. However, there were some cases where no reference values were available or could not be extrapolated due to differences in contrast media or equipment, so I left them blank (especially in contrast-enhanced ultrasound, where there are no reports of kidney evaluation in normal dogs with Sonazoid, as the dose is different from those of other contrast media, and the AUC varies widely depending on the equipment. I also apologize for not being able to include norms in Table 2). Further, I have cited the studies you referred us to.

Reviewer 2 Report

The research consolidates the use of CEUS as a non-invasive method for the evaluation of renal perfusion on the basis of pathogenetic alterations responsible for fibrosis and hypoxia. The paper is well written, clear in its objectives and methodologically well conducted. 

Here are just a few requests for clarification or formal changes:

line 132-133, in which decubitus was the animal placed? Lateral or supine? What does "the measurement area was shortened" mean? Specify that the measurements were made on the left kidney only.

line 152, images A-B are horizontal sections and C-D trasversal ones!

line 165, were recorded up 120 s after.....? 

lines 177-187 and caption of Fig. 3 are redundant! I suggest postponing the explanation of the time-related and intensity-related parameters to Fig. 3 caption, and indicate it in the text.

Table 1, I suggest to horizontally align the parameters to the values, reporting the unit of measure of velocity in the caption.

Author Response

<Reviewer2>

・line 132-133, in which decubitus was the animal placed? Lateral or supine? What does "the measurement area was shortened" mean? Specify that the measurements were made on the left kidney only.

Re1: Thank you for pointing this out. The position is the right lateral recumbent positions. I realized that this was incorrect and have corrected the sentence to indicate that only the left kidney was observed because the dogs were placed in the right lateral recumbent position; however, I deleted the sentence because it was confusing. I also emphasized that only the left kidney was measured.

・line 152, images A-B are horizontal sections and C-D trasversal ones!

Re2: Thank you for pointing this out. I have corrected it.

・text.line 165, were recorded up 120 s after.....? 

Re3: Thank you for pointing this out. I have corrected it.

・lines 177-187 and caption of Fig. 3 are redundant! I suggest postponing the explanation of the time-related and intensity-related parameters to Fig. 3 caption, and indicate it in the text.

Re4: Thank you for pointing out that the contents of the caption of Fig. 3 are explained in the text. We have only explained the abbreviation in the caption of Fig. 3.

・Table 1, I suggest to horizontally align the parameters to the values, reporting the unit of measure of velocity in the caption.

Re5: Thank you for your suggestion. I have created the table as you suggested; however, please note that this has made the table to be very long horizontally and the letters are very small. Also, the layout has been reversed in Tables 2 and 3. I appreciate your suggestion, but please allow me to keep the prototype.

Reviewer 3 Report

The title adequately represents the study.

The abstract section is well written and adequately summarize methodology, results, and significance of the study. However, Authors should indicate statistical analysis applied.

Keywords represent the article adequately.

The introduction section is well written falling within the topic of the study.

The section of Materials and Methods is clear for the reader and it describes well the methods applied in the study. However, Authors should check this section and correct many punctuation errors and some mistakes.

Why Authors did not assess haematochemical parameters (particularly, kidney parameters) of enrolled in dogs before and after the experimental protocol?

Regarding statistical analysis, Did authors check the normal distribution of data by Normality test? Please clarify this aspect.

Results are well presented and discussion section is well written. Authors justified well the main findings with appropriate references.

Authors performed correlation analysis among studied parameters, however, Authors did not discuss the results of these correlation tests. Please focus on this aspect in the discussion section.

Conclusion section should be improved.

Authors should rewrite this section by summarizing the results and the significance of the study.

Tables and figures are nice and well represent the results. Authors should check and standardize the references in the list according to journal guidelines.

Author Response

<Reviewer3>

・The abstract section is well written and adequately summarize methodology, results, and significance of the study. However, Authors should indicate statistical analysis applied.

Re1: Thank you for pointing this out. I have followed your suggestion and added an explanation of the statistical methods.

・The section of Materials and Methods is clear for the reader and it describes well the methods applied in the study. However, Authors should check this section and correct many punctuation errors and some mistakes.

Re2: Thank you for pointing this out. We noticed a few errors. We have corrected all the errors that were found.

・Why Authors did not assess haematochemical parameters (particularly, kidney parameters) of enrolled in dogs before and after the experimental protocol?

Re3: Thank you for your question. We discussed this point; however, the purpose of this study is to investigate real-time renal blood flow changes with epoprostenol in normal dogs. Creatinine, cystatin C, and SDMA are important diagnostic markers in renal dysfunction, but changes are seen only in the presence of renal dysfunction. In dogs with normal renal function, creatinine, cystatin C, and SDMA are not sensitive enough to detect changes in renal function. The most reliable canine renal clearance test also does not provide real-time assessment because it evaluates GFR by collecting blood and urine samples over a period of several hours. Therefore, ultrasound, which can repeatedly capture real-time changes, was used.

・Regarding statistical analysis, Did authors check the normal distribution of data by Normality test? Please clarify this aspect.

Re4: Thank you for your question. We also understand the importance of confirming the normal distribution using the Normality test. However, the number of samples in this study (n=6) is small, and we did not confirm the normal distribution with the Normality test. The reason is that the reliability of any Normality test (Shapiro-Wilk, Kolmogorov-Smirnov, etc.) for this number of samples has been reported to be small by some studies (Razali, N. M., Wah, Y. B.. Power Comparisons of Shapiro-Wilk, Kolmogorov-Smirnov, Lilliefors and Anderson-Darling Tests. Journal of Statistical Modeling and Anlytics. 2011. 2, 21-33.) In fact, even if a small sample size is determined to be normally distributed by the Normality test, it is very likely that the distribution is not nearly normally

・Authors performed correlation analysis among studied parameters, however, Authors did not discuss the results of these correlation tests. Please focus on this aspect in the discussion section.

Re5: This discussion is presented in Lines 337-345 of the text.

Conclusion section should be improved. Authors should rewrite this section by summarizing the results and the significance of the study.

Re6: Thank you for your suggestion. I have rewritten the conclusion according to your suggestion.

・Tables and figures are nice and well represent the results. Authors should check and standardize the references in the list according to journal guidelines.

Re7: Thank you for pointing this out. We have corrected the references according to the guidelines.